# Magnesium-Based Alloys Used in Orthopedic Surgery

**DOI:** 10.3390/ma15031148

**Published:** 2022-02-02

**Authors:** Iulian Antoniac, Marian Miculescu, Veronica Mănescu (Păltânea), Alexandru Stere, Pham Hong Quan, Gheorghe Păltânea, Alina Robu, Kamel Earar

**Affiliations:** 1Faculty of Material Science and Engineering, University Politehnica of Bucharest, 313 Splaiul Independentei, District 6, 060042 Bucharest, Romania; iulian.antoniac@upb.ro (I.A.); a.stere@ortovit.eu (A.S.); hongquan83fpt@gmail.com (P.H.Q.); alina.ivanov@upb.ro (A.R.); 2Academy of Romanian Scientista, 54 Splaiul Independentei, 050094 Bucharest, Romania; 3Faculty of Electrical Engineering, University Politehnica of Bucharest, 313 Splaiul Independentei, District 6, 060042 Bucharest, Romania; gheorghe.paltanea@upb.ro; 4Medical Ortovit Ltd., 8 Miron Costin Street, 011098 Bucharest, Romania; 5Faculty of Medicine and Pharmacy, University Dunarea de Jos Galati, 35 Alexandru Ioan Cuza Street, 800010 Galati, Romania; erar_dr.kamel@yahoo.com

**Keywords:** Mg-based alloys, orthopedy, biomaterials, mechanical properties, clinical translation, biocompatibility

## Abstract

Magnesium (Mg)-based alloys have become an important category of materials that is attracting more and more attention due to their high potential use as orthopedic temporary implants. These alloys are a viable alternative to nondegradable metals implants in orthopedics. In this paper, a detailed overview covering alloy development and manufacturing techniques is described. Further, important attributes for Mg-based alloys involved in orthopedic implants fabrication, physiological and toxicological effects of each alloying element, mechanical properties, osteogenesis, and angiogenesis of Mg are presented. A section detailing the main biocompatible Mg-based alloys, with examples of mechanical properties, degradation behavior, and cytotoxicity tests related to in vitro experiments, is also provided. Special attention is given to animal testing, and the clinical translation is also reviewed, focusing on the main clinical cases that were conducted under human use approval.

## 1. Introduction

Biomaterials are used in different medical applications and are placed in direct or indirect contact with biological environments [1]. One of the most important features of a biomaterial is its biocompatibility, which is defined as the “ability of a material to perform with an appropriate host response in a specific application” [2]. In the past, three generations of biomaterials were developed, which were used in dental implants, knee or hip articular prosthesis, cardiac stents or valves, breast implants, and vascular grafts. Charnley proposed an innovative hip prosthesis, Ridley manufactured the first intra-articular lenses, Hufnagel invented the “ball in a cage” heart valve, and Vorkhees studied the first vascular graft in humans. After 1960, the toxicology and the biocompatibility of a material became very important. The biomaterials that are used today were developed in order to promote responses at molecular and cellular levels. In addition, the prostheses must be integrated and linked to the human body through different physical, chemical, or biological processes (i.e., osteointegration process) [3].

In orthopedy, the most important challenging conditions are related to bone or soft tissue loss due to different types of traumas, cancers, infections, and inflammations. Using the so-called biomaterial scaffolds, the damaged tissues at the surgery place can be reconstructed or modified [4,5,6,7,8,9].

A simple classification of orthopedic implants could be made according to the time they are implanted in the human body. A first class would be the permanent orthopedic implants, which include total joint replacement as knee, hip, elbow, wrist, ankle, shoulder, or finger joints. These permanent devices are designed to be durable and to replace the human parts throughout the patient lifespan [10,11,12,13]. When a specific prosthesis is manufactured, two natural processes, consisting of osteogenic (bone generating) and osteoclastic (bone removing) phenomena [13], must be taken into account. Metallic materials are characterized by high mechanical strength, biocompatibility, and corrosion resistance, but the difference between Young’s modulus of the bone and the metal alloy induces the stress shielding effect. These metal implants require, unfortunately, a second surgery in order to be removed (temporary implants), or sometimes (the permanent prosthesis), revision surgery is needed. In order to address the problem of the temporary implant, biodegradable materials could be used. Firstly, biopolymers, such as polyglycolic acid (PGA) or poly L-lactic acid (PLLA) were used, but due to their low mechanical properties, they can be used only in no load-bearing zones. These types of materials are not adequate for the osteointegration process and can generate inflammatory responses from the host body, thus creating the need for searching different biodegradable metals [14,15].

In this respect, magnesium (Mg) and magnesium alloys, as representative materials, exhibit very good biocompatibility, good mechanical strength, and biodegradation properties. Magnesium has an elastic modulus of about 45 GPa, a value close to that of the bone (15–25 GPa), resulting in a hindered stress shielding phenomenon. Mg alloys density is comprised between 1.74 and 1.84 g/cm^3^, depending on the alloying components being almost equal to the one of bone (1.8–2.1 g/cm^3^).

Unfortunately, Mg has a low standard electrode potential of −2.372 V, with an increased tendency to corrosion in aqueous media [16,17]. A great advantage of Mg consists of the fact that its degradation products are not toxic and they even promote new bone formation [18,19]. The different fracture types in human and animal models resolved with temporary fixation devices made by magnesium alloys are presented in Table 1.

Different unwanted reactions are presented, with some studies [30,31] showing that aluminum, as an alloying element (Al), could be linked to neurons and osteoblast decrease in the human body, leading to Alzheimer’s disease or dementia. In addition, rare earths can cause hepatic toxicity [32]. Mg-RE as Mg-Y-RE-Zr (MAGNEZIX^®^-Syntellix AG, Hanover, Germany) was officially accepted in 2013 as temporary implant material and received the CE (Conformité Européen) mark. In the case of the abovementioned material, the degradation time for in vivo applications is longer than in the case of other Mg alloys. The implants can maintain their mechanical properties, so that the tissue has time to heal, and the degradation products are released in a lower quantity. They are sold as bioabsorbable screws, pins, and wires [33]. The Mg screws are met in a large variety of forms, and they are usually indicated in small bone fractures and arthrodesis, including scaphoid fractures, medial and lateral malleolar fractures, avulsion fractures, intra-articular fractures of the tarsals, osteotomies around the foot and ankle, and the fracture of patella and ulna. Another magnesium-based alloy that is commercially used is Mg-Ca-Zn. This material was approved for clinical trials by Korea Food and Drug Administration (KFDA) in 2015, in order to treat hand fractures.

Usually, Mg is alloyed with zinc (Zn), manganese (Mn), calcium (Ca), silver (Ag), zirconium (Zr), rare earths (RE), and yttrium (Y), resulting in six binary alloys as follows: Mg-Y, Mg-Zn, Mg-Ca, Mg-Ag, Mg-Zr, and Mg-RE (Figure 1). The alloying process is a general method used to improve the pure magnesium properties [34].

Zinc is often used as an alloying element for Mg because it decreases the nickel (Ni) and iron (Fe) impurities’ corrosive effects. Its concentration is limited at about 3%. In addition, Zn is considered an essential element, as Zn deficiency can result in severe disruption of physiological functions in the human body. Manganese is mainly used to increase the ductility of the material. In a lower quantity, Mn contributes to mechanical strength improvement. It can also improve the alloy viscosity when used in a quantity lower than 2 wt.%. Calcium helps to increase the mechanical resistance of solid solutions and precipitates. To a large extent, it can also act as a grain refining agent and contribute to grain boundary strengthening. In Mg-Ca binary alloys, Laves-type intermetallic phases with the composition of Mg_2_Ca are formed, having the creep resistance increase as a result [35]. Zirconium is commonly used as a grain refining agent and to increase the mechanical strength in Mg alloys. It also reduces the effect of Fe impurities on the corrosion resistance of Mg alloys. Rare earths (RE) have an important contribution to the mechanical strength of the material. Yttrium has a high solubility in Mg, being the most used RE alloying element, although neodymium (Nd) and cerium (Ce) are also used [36].

The fabrication methods used for Mg-based implants’ have a direct influence on their properties and performance. The main conventional techniques used at the industrial level are casting, powder metallurgy, and laser-based additive manufacturing technology.

Casting is one of the easiest and cheapest technology, but some variations of this method are common: squeeze casting, high pressure gravity casting, sand casting, and stir casting [37]. The unwanted segregation phenomenon in the as-cast material can be corrected by heat treatment or hot processing.

The powder metallurgy route has a high precision in Mg-based implants manufacture. The main methods through which the Mg powders are made are the atomization of molten metal, electrolysis, evaporation-condensation, and mechanical crushing. [38].

Laser-based additive manufacturing technology is still a new method for Mg-based implant manufacture. By this technology, the final product is printed layer-by-layer, based on computer-aided tomography [39,40]. This is the most used technique, which helps doctors and engineers to develop custom-made Mg-based implants [41,42]. However, because Mg is characterized by a low boiling point and a high vapor pressure, it can be burnt during laser processing [43]; therefore, it must be produced using a high-vacuum chamber or a protective environment. It is still an actual challenge to develop such a material and to choose a proper production method; therefore, the Mg-based alloy exhibits ideal properties and is successfully used in orthopedic surgery [44,45].

Further, we present the main attributes of Mg alloys for temporary orthopedic implants, the current status of Mg alloys used in orthopedic surgery, different studies performed on animal models, and the clinical translation for Mg-based materials for manufacturing biodegradable implants for orthopedic surgery.

## 2. Attributes of Mg Alloys for Temporary Orthopedic Implants

Biodegradable metals are a new paradigm for orthopedic implants because they promote bone formation [46] and sustain the bone healing and remodeling process through a gradual load transfer between implant and tissues [47]. In addition, they are designed to provide sufficient mechanical strength at the beginning of the treatment, and then, after their degradation, a complete bone healing process is observed [48].

The key features for biodegradable Mg alloys, suitable for temporary orthopedic implants, are biocompatibility, proper mechanical properties to assure mechanical integrity until the fracture healing, degradation rate, and dynamic corrosion (some authors use the term flow rate), according to the clinical needs [44,45,46,47,48]. The effective biofunctionality of the biodegradable temporary orthopedic implants can be evaluated only by in vivo testing, on animal models, followed by clinical trials. In addition, the new bone formation, bone–implant interface, and inflammatory reactions can be evaluated. The methods used for the evaluation of the biodegradable Mg alloys for temporary orthopedic implants are shown in Figure 2.

Magnesium can be found in the human bone [16], ligaments and tissues [49], and along with other substances, in body fluids [50]. The products that are generated due to Mg degradation are absorbed by the macrophage cells and then eliminated through the renal route [50]. In Figure 3, the Mg absorption phenomenon and excretion equilibrium in the human body are presented.

During degradation processes, the Mg ions have a positive impact on stem cells. Abed et al. [51] studied and proved that Mg ions are present in the extracellular matrix. Other studies [52] showed that Mg-ions can increase osteoblasts cells viability or have an important role in gap junction intercellular communication (GJIC) between material and osteoblast cells and signal transmission, with all of the investigations showing good biocompatibility of the magnesium implants and a sustained bone remodeling process [53,54]. It was underlined that the magnesium ions could have a detrimental effect on implant surrounding cells and tissues, and they may cause a systemic toxicity if they are present in a high amount [55]. Different cellular lines are cytocompatible with Mg-based alloys such as MC3T3-E1 murine osteoblasts, MG-63 human osteosarcoma cells, mouse fibroblast, RAW264.7 macrophages, and primary human mesenchymal stem cells [56]. The Al and Zn percent in AZ31 alloy is considered to be in the safe range for in vivo applications (1% for Zn and 2–3% for Al), and it does not exhibit toxic behavior inside the human body [57,58], but Witte, Antoniac, and other authors recommend avoiding magnesium alloys that contain aluminum because of their neurotoxicological characteristics [46,55,59]. In Table 2 and Table 3, the physiological and toxicological characteristics of the alloying elements (described in the Section 1—see Figure 1) and of the impurities that are usually met in Mg-based alloys are presented. Table 4 presents the chemical composition of some biodegradable magnesium alloys for medical applications.

The mechanical integrity of magnesium alloy implants is, as noted before, a key factor in orthopedic applications [60]. The fracture toughness of this material is higher than that of ceramic materials. A disadvantage of the magnesium alloys is the low value of compressive yield strength, ranging between 65 and 100 MPa, by comparing it with that of the human bone (130–180 MPa) [61,62]. A studied pure Mg implant [63] for osteonecrosis of the femoral head proves that bone flap stability was increased, but unfortunately, two patients suffered the collapse of the femoral head, a fact that sustains that pure Mg is not suitable for load-bearing biomedical applications. Further improvements, such as alloying Mg with different biocompatible materials, have to be made for Mg-based implants in order to be used in load-bearing zones of the human or animal models’ bodies [61].

Usually, the enhancement of the mechanical properties is made through grain refinement, by adding rare earths elements (Ce, Y) or alloying elements such as Zr, Sr, Ca, and Zn [64]. The typical Mg-based alloys that contain aluminum (Al), using the ASTM alphanumeric designation system, are AZ91, AZ31, AE21, and LAE442. The Mg-based alloys that do not include Al in their chemical composition are WE, MZ, WZ, and Mg-Ca type binary alloys. The WE43 alloy contains Y, Zr, and RE, and it exhibits a very good creep resistance and stability of the mechanical properties at high temperatures. In industrial practice, the Mg-based alloys have an increased hardness when rare earth metals are used [65]. In Table 5, the mechanical properties of Mg-based alloys, which do not contain aluminum, are presented.

Mg alloys designed for medical implants are degraded in physiological environments. One of the problems that must be solved is the rapid corrosion process of Mg, which leads to gas cavities formation and an important decrease in material mechanical strength. Figure 4 schematically represents the degradation behavior of Mg-based temporary bone implants, in correlation with the bone fracture healing process, in ideal conditions.

By galvanic corrosion process, an accumulation of magnesium hydroxide appears on the Mg alloys surface that acts as a corrosion protective layer. This layer exhibits a porous structure, which can be affected in chlorine ions solutions. This process also happens inside the human body, due to the high quantity of chlorine ions and because the human fluids accelerate the Mg degradation. It was found that other factors, such as organic buffering molecules, inorganic ions, or dissolved oxygen species, have an influence on Mg corrosion [75].

An altered degradation behavior, due to mechanical tension and compression, was shown in [64]. The place where the implant is used can also influence the Mg degradation process. Chaya et al. [26] observed an important corrosion rate in the case of a Mg plate, which was implanted in a loaded ulna fracture model. In this case, plates are surrounded by muscles with a higher amount of water and blood that accelerate the corrosion process, compared to a bone implanted Mg screw (no fluids around) that exhibits different behavior.

Most of the papers from the literature report the testing of Mg biodegradable alloys by immersion, hydrogen evolution, and electrochemical testing methods, in different testing mediums. According to Sekar [62], to Mei [75], and to other authors [47,55,61] that have made critical reviews on selecting mediums for corrosion testing of biodegradable metals, the commonly used corrosive media for evaluating the corrosion rates are NaCl solution, complex saline solutions, simulated body fluids, cell culture mediums, and protein-containing solutions. Typically, the corrosion rate of magnesium is decreased as a function of the increasing complexity of the media. In addition, it is well established that corrosion rates obtained in vitro do not match with the in vivo studies [31,53].

A schematic illustration of the corrosion behavior of Mg in the commonly used media is shown in Figure 5.

The regulation of degradation processes (Figure 6) must be taken into consideration for Mg-based implants, based on physical or chemical methods. One important issue is to point out that the alloying element and metallurgical processing, such as heat treatments and plastic deformation, strongly influence the biocompatibility, the mechanical properties, and the degradation rate, due to the modifications induced in the microstructure of magnesium alloys [76,77]. Based on this consideration, the microstructural characterization of the biodegradable Mg alloys by optical microscopy and scanning electron microscopy, coupled with EDS, appears to be mandatory in order to understand the microstructural features that influence the biofunctional properties. All these analyses on the microstructure of the biodegradable Mg alloys support researchers in understanding and modulating the biodegradation process.

In Figure 7, we present some examples of different microstructures on biodegradable Mg alloys, which were obtained by our research group. In addition, various surface treatments or coatings could influence the degradation rate, as well as the interaction with surrounding tissues. Although these aspects are not covered in this review, we consider it relevant to mention some minimal details.

Surface modifications are useful mainly for adapting the corrosion process to the clinical needs for biodegradable Mg alloys. Some of the most important surface modifications are self-passivation of the Mg, hydrothermal treatment, and alkaline heat treatment. [34]. Hydrothermal treatment is a surface treatment made by using sodium hydroxide (NaOH) or deionized water that assures a compact and uniform layer of Mg(OH)_2_. The layer thickness and morphological properties were found to be strongly dependent on the medium pH and hydrothermal treatment time. Other types of surface modifications are the chemical ones that include chemical passivation, reaction with ionic liquids, self-assembled monolayers, chemical conversion coating (phosphate or fluoride conversion coatings on Mg), and sol-gel coating technology (titania, hydroxyapatite, bioactive glass, and polymer coating) [71]. Chemical conversion coating is another technology that hinders Mg-based alloys to corrode. Fluoride conversion coatings are biocompatible, determine a gradual degradability of Mg alloys, and sustain calcium phosphate deposition. The release of fluoride ions from the MgF_2_ does not have a cytotoxic effect. It exhibits antibacterial properties and is biocompatible. Bioactive glass coating is a proper technology because it is usually used in tissue engineering and exhibits high bioactivity, controllable biodegradability, and good osteoconductivity [62]. Other technologies could be considered as silane coatings and biomolecules coatings. Biodegradable polymer coatings are an important class, which includes coatings with poly(-caprolactone) (PCL), poly(L-lactic acid) (PLLA), and poly(glycolic acid) [61]. It has been proven that they reduce in a large amount the corrosion rate of Mg. Natural polymer coatings such as collagen, stearic acid, chitosan, and serum albumin should also be taken into account because the compounds exhibit a biomimetic nature [43].

From the electrochemical surface modification class, it is important to mention the anodizing and microarc oxidation methods [77]. Microarc oxidation or plasma electrolytic oxidation (PEO) are more and more used, in order to control the corrosion resistance of Mg alloy. Another electrochemical surface modification is the electrodeposition that is based on a pulsed current mode, and high-quality coatings are obtained.

There are also physical surface modifications methods, namely physical vapor deposition, ion implantation, sputtering, and laser surface modifications such as laser surface melting and laser surface alloying. Of the plasma surface modification technologies, the most used are physical vapor deposition, ion implantation, and plasma spraying [61].

Laser treatments are made to change the surface of Mg alloys without affecting the bulk material properties. After this technology is applied solid solutions on the metal surface are formed, because laser treatment is associated with a high cooling rate. This method presents numerous advantages such as a complex geometry treatment, and it does not require a vacuum chamber.

From the mechanical surface treatments, the surface mechanical attrition is a very interesting procedure because it induces a plastic deformation, which is beneficial to the Mg grain refinement that improves the mechanical properties of the alloy. Other mechanical treatments are friction stir processing, severe plastic deformation, abrasive water jet machining, hybrid dry cutting-finish burnishing, and staggered extrusion.

Osteogenesis consists of new bone tissues development in order to repair a fractured bone [78]. This process is necessary for a material to work as a temporary implant. Different biochemical and pathological factors have an influence on osteogenesis. Bioactive materials can be used as coatings layers on the magnesium-based implants to increase this physiological process. Osseous growth proved to be an important property of Mg implants, and it was shown using many animal models. The implant design consisting of a plate/screw system was used for small pigs, beagle dogs, or rabbits. In the case of rodents, the intramedullary fixation with pins, rods, and bars was used. An effective fracture treatment has to be focused not only on the broken bone reconstruction but also on improving the bone quality through the osteogenesis process. The preclinical studies prove that Mg-based alloys are a good choice when the animal fracture is not placed in a load-bearing zone, and a good bone formation was shown using medical images [53].

Agarwal et al. [79] studied the angiogenesis that includes new blood vessel formation, on fetal mouse metatarsal assay. The influence of Mg alloys was investigated through a protein lysate using a cytokine array consisting of angiogenic activators and inhibitors. It proves that angiogenesis is beneficial to bone-healing, Mg-based temporary implants helping this process to a high extent.

## 3. Current Status of Mg Alloys for Orthopedic Applications

The main Mg-based alloys used or tested for orthopedic applications are presented in this section. Aspects regarding chemical composition, microstructure, mechanical properties, biodegradability, and biocompatibility of different Mg-based alloys are described.

### 3.1. Mg–RE-Based Alloys

Rare earths (RE) are alloying elements that determine an increase in the mechanical strength, an improvement of the corrosion resistance, and a high creep resistance of the Mg. Yttrium (Y), gadolinium (Gd), cerium (Ce), and neodymium (Nd) are the most used rare earth elements in the case of Mg-based alloys designed for medical applications. Better mechanical properties were observed after using a combination of Zn or/and Zr with RE elements, which are present in the commercial WE43 alloy [69,80]. The formation of an intermetallic phase in the as-cast or extruded Mg-RE alloys is due to the high maximum solubility limit of rare earth elements [81]. Dobatkin et al. reported that Mg-4.7Y-4.6Gd-0.3Zr (all in wt.%) show, for the as-cast state, the formation of α-Mg matrix and Mg_24_(YGd)_5_ precipitates, with the mechanical properties being improved by hot extrusion [66]. The Mg-RE alloys exhibit a good degradation behavior [74,82]. Gu et al. performed cytotoxicity tests on different binary Mg-based alloys that also include analysis on Mg-Y alloys and concluded that Y exhibits some toxic effects on different cell lines [83].

### 3.2. Mg–Zn-Based Alloys

Mg-Zn binary-based alloys have an α-Mg matrix and MgZn precipitates [84]. In an as-cast state, by alloying with 4–5 wt.% Zn results in an increase in mechanical strength, but higher Zn content, drastically leads to the properties deterioration [57]. Heat treatments, which increase the solubility of Zn in Mg, can increase the percentage of Zn up to 6.2–9 wt.% [85]. Ternary or quaternary alloy systems, with either Ca, Zr, Sr, Mn, or Y, in association with Mn-Zn-based materials have shown good mechanical properties.

Alloys from the ternary system Mg-Zn-Ca are characterized by an improved corrosion resistance and increased strength [86]. Calcium is considered to be a grain refining element [87], but it must be limited to 0.2 wt.%. [88]. Mg-Zn-RE alloys are a ternary important class. However, introducing an RE as a primary alloying element in concentrations above 5 wt.% can induce a severe effect of toxicity in the human body. RE improves the mechanical properties. In the case of Mg-8Zn-1.6Y, in as-cast state, the Mg_7_Zn_13_ and Mg_2_Zn_3_Y_3_ phases are observed, which are known to produce an increase in the mechanical strength [89,90]. The mechanical strength of Mg-Zn-RE alloys is higher than that measured in the case of commercial Ti, and it could be explained on the basis of very fine grain size, widely dispersed in a hard lamellar phase [91]. In the ternary system Mg-Zn-Zr, Zr act as an important grain refiner [92]. The alloys such as Mg-3Zn-0.6Zr (ZK30) and Mg-6Zn-0.6Zr (ZK60) exhibit good mechanical properties as follows: for ZK60 (YS = 235 MPa, UTS = 315 MPa and elongation rate of 8%), and in the case of ZK30 (YS = 215 MPa, UTS = 300 MPa and elongation rate of 9%) [93,94]. The main explanation of the mechanical properties’ improvement is related to the microstructural modification induced by the presence of Y that generates an additional phase-Mg_2_Zn_3_Y_3_ into the alloy’s microstructure.

The degradation behavior of binary Mg-Zn alloy was analyzed by Nanda et al. [95], where five Mg-xZn alloys, with x = 2, 4, 6, 8, and 10, were tested. A direct proportionality between Zn content and a more positive value for the corrosion potential was observed. The enhanced degradation process could be explained on the basis of MgZn phase formation in the alloy matrix, which acts as a barrier against ions diffusion, reducing the electrochemical reaction of the alloy and the electrolyte. The binary alloy Mg-6Zn presents the best degradation behavior (E_corr_(V) = −1.67, i_corr_ (μA) = 122 and corrosion rate (mm/year) of 2.78), by comparing to those obtained for Mg-2Zn (E_corr_(V) = −1.86, i_corr_ (μA) = 210, and corrosion rate (mm/year) of 4.8) and Mg-10Zn (E_corr_(V) = −1.74, i_corr_ (μA) = 135 and corrosion rate (mm/year) of 3.08).

Conclusions from the literature show that an increase in Zn content over 6 wt.% can form other intermetallic phases in Mg-Zn alloys that determine an increase in the corrosion process in these binary alloys [95]. In Mg-Zn-Ca ternary alloys, the best corrosion properties are reported to be for the Mg-4Zn-0.5Ca alloy after testing in vitro and in vivo, mainly due to the ternary Ca_2_Mg_6_Zn_3_ phase formation in the α-Mg matrix. An increase in Ca content above 0.5 wt.% leads to a course Mg_2_Ca phase along the grain boundaries, founded also in Mg-Ca alloys, which acts as anodes in the galvanic coupling, with the α-Mg phase making the alloy prone to high galvanic corrosion. In the case of Mn-4Zn-0.2Ca, the corrosion rates were reported to range between (2.43 ÷ 2.67) × 10^−4^ A/cm^2^ compared to the pure magnesium that is 3.71 × 10^−4^ A/cm^2^ [96]. In vitro and in vivo studies reported a 35 ÷ 38% degradation rate that took place in the first 3 months [97]. Excellent corrosion resistance is present in the case of Mg-Zn-RE alloys such as Mg-Zn-Gd and Mg-Zn-Y. Mg-1.8Zn-0.2Gd presents a relatively low corrosion rate (<0.28 mm/year) after in vitro studies, and for in vivo analysis, the structural integrity was maintained for two months, with almost complete dissolution after 6 months.

An improved configuration was reported by Liu et al. for Mg-2.4Zn-0.8Gd with a 0.21 mm/year corrosion rate [81]. For Mg-Zn-Y alloys, immersion tests in Hanks’ solution show that alloys with Mg_3_Zn_6_Y 2^nd^ phase exhibit a very low corrosion rate (<0.1 mg/cm^2^/h) [98]. For Mg-Zn-Zr ternary alloys with approximately 0.5Zr, a corrosion rate of 0.006 mg/cm^2^/h in as-extruded and as-cast state was observed. After 24 h, the extruded alloy presents a lower degradation rate, while the casted samples continue to have similar behavior. This fact is associated with the formation of a HA layer on the extruded layer [99]. In the case of the Mg-Zn binary alloy, it was concluded that cell morphologies in different extracts are normal and healthy, by comparing them with the negative control [57]. According to ISO 10993-5: 1999, the cytotoxicity of these extracts was evaluated at grade 0–1, a fact that shows that Mg-6Zn alloy is suitable for cellular applications. For Mg-4Zn-0.2Ca, mouse osteoblast cells were adopted to evaluate their cytotoxicity. For the 100% as-extruded and as-cast ZK60 extracts, significant toxicity was shown with a cell number reduction of 40% [99].

### 3.3. Mg–Ca-Based Alloys

Calcium ions exhibit a benefic effect on the bone-healing process [100,101]. The binary Mg-Ca alloy contains an α-Mg matrix and Mg_2_Ca as the second phase [102,103]. This second phase is placed at the grain boundaries, generating grain boundary pinning and grain refining and increasing the material strength. Ca presents a limited solubility in Mg. Adding Ca in a higher amount than 1%, the ductility of the alloy, due to Mg_2_Ca phase formation, can decrease [104]. The as-cast Mg-1Ca alloy has unsatisfactory mechanical properties; therefore, hot rolling or hot extrusion is necessary for grain refining [105]. If Zn is added to the Mg-Ca alloys, an important improvement of the mechanical properties can be noticed [106]. After solution treatment, quenching, and age hardening of Mg-Ca-Zn alloy, the Mg_2_Ca phase is dissolved and, finally, results in Ca_2_Mg_6_Zn_3_ phase formation inside the grains. Adding Sr to Mg-Ca alloys in order to obtain Mg-Ca-Sr alloys has shown mechanical properties enhancement and beneficial effects to the osteogenesis process [107]. Fernandes et al. [108] investigated the mechanical strength of the Mg-Ca-RE alloys, respectively Mg2Ca2Gd and Mg1Ca2Nd (wt.%), in a preclinical study on New Zealand rabbits and conclude that the addition of 2%wt Gd has an important effect on grain size reduction (from 190 to 51 μm) that generates higher mechanical properties. The degradation behavior of Mg-0.8Ca was investigated by Mohamed et al. [109]. They showed that pure Mg has a higher tendency to passivate compared to Mg-0.8Ca binary alloy, a fact sustained by the obtained values for E_corr_ (−1.73 ± 0.01 V), i_corr_ (0.05 ± 0.01 mA/cm^2^), and corrosion rate (1.08 ± 0.38 mm/year). These values are higher than those obtained in the case of pure Mg (E_corr_(V) = −1.59 ± 0.05, i_corr_ (mA/cm^2^) = 0.02 ± 0.01 and corrosion rate (mm/year) of 0.35 ± 0.17). Bita et al., Chen et al., and Liu et al., showed that after immersion in SBF of Mg-Ca alloys, the corrosion products that are formed on the material surface are mainly Mg(OH)_2_, MgO, CaO, and Ca(OH)_2_ [110,111,112]. The SBF solution contains chloride ions, which deteriorate the electric double layer formed on the Mg-Ca surface, and it electrochemically reacts with the alloy matrix and causes pitting corrosion [113].

### 3.4. Mg–Zr-Based Alloys

Zirconium, as an alloying element, promotes osteointegration and is characterized by high biocompatibility and low ionic cytotoxicity. The maximum solubility percent of Zr in Mg is about 3.8 wt.%, and the addition of Zr leads to important grain refinement. For Mg-xZr-ySr (with concentration values for x and y lower than 5%), good mechanical integrity and high corrosion resistance were observed. The Mg-1Zr-2Sr presents an ultimate compressive strength of 230 MPa and a 31% compressive strain. If holmium (Ho) is added, intermetallic phases such as MgHo_3_ and Mg_2_Ho are formed. Together with Mg_17_Sr_2_, they contribute to the increase in the ultimate compressive strength (UCS) to 250 MPa and the compressive strain to 32% [114]. Another important ternary alloy is Mg-1Zr-1Ca in as-cast condition, which has an UCS of 175 MPa, and after hot rolling, this value is raised to approximately 300 MPa [115]. The degradation behavior of Mg-xZr-ySr, with *x* and *y* equal to 1%, 2%, or 5%, was studied by Li et al. [73]. In the Mg alloy with 1%Zr, the material with 5%Sr has the highest value of i_corr_ (3 × 10^−3^ A/cm^2^) in SBF, while the alloy with 2% Sr shows the lowest value (0.5 × 10^−3^ A/cm^2^) in SBF. The alloys with 2% and 5% Zr present a similar variation for different Sr content (e.g., for Mg-2Zr-ySr: i_corr_ = 2.5 × 10^−3^ A/cm^2^ in SBF, for Mg-5Zr-ySr: i_corr_ = 6 × 10^−3^ A/cm^2^ in SBF, where y = 2% or 5%) [73]. The cytotoxicity and osteoinduction capacities of Mg-xZr-ySr were investigated showing, in the case of Mg-1Zr-2Sr and Mg-2Zr-5Sr, that these materials promote initial cell attachment and growth [73].

### 3.5. Mg–Sr-Based Alloys

Mg-Sr binary alloy, in the as-cast state, has a dendritic structure and consists of an α-Mg matrix and a Mg_17_Sr_2_ intermetallic phase that precipitates along the dendritic arms [116]. When the Sr concentration increases, the mechanical properties improve, due to the dispersion of the second phase precipitates.

The corrosion rate of the as-rolled Mg-xSr (x = 1%, 2%, 3%, and 4%) alloys was determined, based on weight loss, hydrogen evolution volume and potentiodynamic polarization measurements [117]. In the case of the last procedure, a decrease in corrosion rate up to 2% Sr was noticed, followed by an increase in corrosion speed for 3% and 4% Sr content. The hydrogen evolution and weight loss determinations showed similar corrosion rates. Strontium has shown a positive effect on osteoblast cell growth and new bone formation in parallel with a decrease in bone resorption.

The cytotoxicity of Mg-xSr was analyzed by the indirect contact method between MG63 cellular line and as-extruded Mg-Sr alloys. It could be observed that at 2, 4, and 6 days, the cells cultured in Mg-0.5Sr extract, presented a higher absorbance rate than the control ones at each essay. The MG63 morphology was normal, and it was concluded that Mg-Sr alloy is a biocompatible material [117].

### 3.6. Mg–Ag-Based Alloys

Binary Mg-Ag was proposed as an implant material, due to the biodegradability of Mg and antibacterial properties of Ag. In such types of materials, the β phase (Mg_4_Ag) is present. After a heat treatment applied for Mg-2Ag, Mg-4Ag, and Mg-6Ag, the mechanical properties of the alloys have improved. The UTS in the case of cast Mg-6Ag was found to be equal to 215.9 ± 11.3 MPa, and the UCS was about 244 ± 9.2 MPa. The Ag presence determines a notable improvement in the material ductility, double when comparing it to that of pure magnesium [118,119].

When the Ag content increases, the binary alloy Mg-Ag can easily corrode regardless of the applied heat treatment. The lowest degradation rate was noticed in the case of Mg-2Ag. A corrosion rate of 0.343 mm/year was measured, a value which is much lower than in the case of pure magnesium (0.534 mm/year) [118].

To analyze the cytotoxicity and cytocompatibility of the Mg-Ag binary alloys, long-term cytotoxicity tests, and cell adhesion tests over 14 days, were made. Human osteoblasts cells were developed directly on the alloy sample. In conclusion, Ag has a benefic effect on corrosion rate decrease, without supplementary induced toxicity, and it generates a better medium for cell adhesion.

## 4. Animal Testing

The preclinical research is important for Mg-based implants’ manufacture, although there are some variations in the chosen animal models and fracture types. Usually, animal models such as rabbits, rats, miniature pigs, beagle dogs, engineered mice, and goats are involved. In some studies, which include rats or mice, many more animals than in analysis that involve bigger species such as dogs or goats are used [120]. In animal model testing, there are no adopted standard procedures regarding their size and the implant location. The selection of a specific model type is based on implant application, and it must be considered that the water content and blood flow rate depend on the overall size of the animal. Usually, small animals are used for material testing and large animals for specific geometry implant testing (Figure 8).

In Table 6, some examples from the literature with different animal models and Mg-based implants involved in studies are presented.

Mg-RE alloys were generally evaluated in vivo on small- to medium-size animal models. Chow et al. [124] investigated the safety and efficacy of WZ42 magnesium alloy, compared to nondegradable Ti6Al4V, on SD rats. The implanted samples were intramedullary pins of 15 mm length × 1.66 mm diameter, used to fix a full osteotomy on femoral bone and wires of 20 mm length × 0.68 mm diameter, which were wrapped around the femur. It was found that the intramedullary pins were subjected to stress-related corrosion effects due to high mechanical loading and to surrounding vascular corrosion, which determined the implant failure. The biocompatibility of the WZ42 was found in natural limits with no accumulation of Mg or other alloying elements in animal bodies. The histological analysis shows a normal fracture healing at the implant site, indicating that WZ42 must be considered as a potential Mg-based alloy used in the orthopedic field in low or medium load-bearing sites.

The WE43 alloy function on osteosynthesis was analyzed by Marukawa et al. [130], considering a bone fracture model of a beagle dog tibia. Screws of 13 mm length and 2.6 mm diameter were prepared using monolithic and anodized WE43 and implanted in the tibia region. It was shown that both types of Mg alloy implants do not exhibit systemic inflammatory reactions on the fracture site, and the anodizing treatment determines a protective effect of the screw against excessive body fluid corrosion. In addition, at the fracture site, increased mineralized bone area and mineral apposition rate, in the case of anodized WE43 screw, were observed. This type of alloy shows good biomechanical properties for load-bearing sites. Naujokat et al. [20] analyzed the same type of alloy (WE43) in four-hole plates (1 mm thickness, 22 mm length) and in cortical bone screws (2 mm diameter, 5 mm length) for cranial fractures of minipig animal models. The implants were manufactured by powder metallurgy followed by hot extrusion. It was noticed that the bone placed in the implant neighborhood was affected by lacunas’ formation. Although the nature of the lacunas’ generation process is not entirely understood, the entire procedure leads to undisturbed bone healing in all investigated cases. Imwinkelried et al. [132] underline the effect of plasma electrolytic coating on the strength retention of degraded Mg implants. Using minipigs as animal models, they have prepared rectangular plates of 60 × 60 × 1.5 mm^3^, made from a modified WE43 alloy, with a lower impurity level. It was found that the in vivo degradation of the Mg alloy was four times slower than the degradation obtained when the devices were immersed in SBF solution. The coated implants exhibited higher strength retention compared to the uncoated ones, and the maximum value was attained after 12 weeks. This result is in good agreement with the anatomical time needed for a fracture to be healed. Bita et al. [134] studied magnesium-based rivet-screws, manufactured from a modified WE43 allot with a 10 μm plasma electrolytic coating or uncoated, implanted in order to fix a mandibular defect, in mini pigs. The rivet screws were tubular implants with threads and an outer diameter of 2.43 and 2.53 mm, an inner diameter of 2.1 and 2.2 mm, a length of 6 mm, and a thickness of 0.165 mm. The degradation of the implants was slow, and a plastic deformation was noticed during the rivets’ activation step. The surface coating proved a benefic effect on the Mg degradation rate, and an improved bone density around the implant zone was observed.

Another rare earth element used in Mg-based alloys is Nd. The promising properties of this type of alloy were analyzed in the big-size animal model (goat) and improved using a brushite coating (known as JDBM–DCPD) [131]. Femoral screws of 45 mm in length and 4.5 mm in diameter were implanted as a fixation device for a manmade defect and analyzed at an 18 month interval. The coated screws present a superior osteoinductivity and slow degradation rate compared to the pure JDBM alloy. The overall degradation was not as satisfactory as expected.

Mg-Zn-Ca alloy (ZX00) was used for bone fracture stabilization in a sheep animal model by Holweg et al. [25]. Proximal screws with an outer diameter of 3.5 mm and a length of 29 mm, with an initial volume of 198.4 mm^3^ and a surface area of 359.8 mm^2^, were inserted into the tibial shaft of the animal model. The bone defect created by osteotomy had a diameter of 359.8 mm, a length of 24 mm, and a volume of 173.6 mm^3^. Three weeks after implantation, an initial degradation of 8.7% of the volume, was observed. Between week 3 and 6, the volume of the implant remains unchanged, and after 12 weeks, the screw volume was significantly reduced, determined to be 180.7 ± 10.2 mm^3^. The main conclusion is that decreasing the Zn content and balancing the Ca content result in an alloy with high mechanical strength and low corrosion rate.

Mg-Ca alloys show promising qualities in orthopedic implantology [46,104,105,113,122,129,134]. A preclinical analysis was conducted by Neacsu et al. [122] on an albino rat animal model. They used uncoated and coated with cellulose acetate (CA) Mg-1Ca-0.2Mn-0.6Zr alloy, from which intramedullary bars with an area of 2 mm × 16 mm, were prepared. The fracture of the femur was made in the middle third of the bone, and the implants were introduced in the intramedullary channel. In the left femur, uncoated material was implanted, and in the right femur, the cellulose acetate coated nail. The histological sections from the rat bones implanted with coated or uncoated Mg-based alloys presented structural alterations, with fractured bone lamellae and lost cellular details. Peri-implant fibrosis was presented, and new bone formation was also observed. In the case of uncoated implants, regeneration was mainly due to scar formation. The biocompatibility of the analyzed alloy was good, and for CA-coated implants, less bone destruction and mild to moderate fibrosis were detected. Chen et al. [111] made in vivo studies on Mg-Ca and Mg-Ca-Zn-Ag alloys using cylindrical samples with a diameter of 2.5 mm and a length of 4 mm implanted in osteoporotic SD rats’ femoral defects. All investigated alloys exhibited a similar cell morphology and proliferation, without any inflammatory reactions. After 1 week of implantation, Mg-Ca-Zn-Ag alloys presented a better integration in the bone tissue compared to Mg-Ca alloy. The alloy Mg-0.8Ca-5Zn-1.5Ag shows higher osteogenic activity and bone substitution rate, because of the increased corrosion resistance and better release of metallic ions with a stimulatory effect on the fracture site, leading to a gain in bone volume and high-quality new bone formation.

Antoniac et al. [129] investigated the Mg-1Ca alloy as suitable materials for applications in small bone fracture repair. Samples were implanted into the greater femoral trochanter of the Oryctolagus Cuniculus rabbit animal model. It was noticed that local tissue metabolism influences the corrosion process, and the Mg-1Ca alloy has all the biocompatibility requirements. During a short-term observation of the rabbits, using X-ray, it was clear that the investigated material does not generate local toxic products or gas bubbles after implantation in bone.

Li et al. [73] studied the Mg-Zr and Mg-Zr-Sr alloys for their potential as biodegradable implant material. Using cylindrical samples with a diameter of 2 mm and a length of 4 mm made from different alloys Mg-5Zr, Mg-1Zr-2Sr, and Mg-2Zr-5Sr, they perform in vivo studies on NZW rabbits, by implanting the samples in the cortical region of the femur. It was concluded that all investigated alloys could induce new bone formation around the implantation site and the Sr addition increases the osteointegrative properties of the alloys. Having good mechanical properties and excellent biodegradation behavior the alloy Mg-1Zr-2Sr could be considered some of the best candidates as an implant material for orthopedic applications.

Gu et al. [117] analyzed a Mg-2Sr binary alloy produced by the rolling technique, which exhibits high mechanical strength and low corrosion rate. This material is characterized by a Grade I cytotoxicity, and it induces a high alkaline phosphatase activity. Cylindrical rods with a diameter of 0.7 mm and a length of 5 mm were made from as-rolled plates as intramedullary nails for implantation in mice (C57BL/6). The corrosion of Mg-2Sr was dependent on the implantation region. It was noticed that faster corrosion appears in the distal zone (the metaphyseal region with trabecular bone), by comparing it with the results obtained in the proximal femur region. The degradation of the implants simultaneously happened with new bone formation, seen in the first 2 weeks. After 4 weeks, new-formed bone is integrated, showing a smooth surface, and the peri-implant cortical bone presents an increased thickness.

Jähn et al. [121] proposed the use of binary alloy Mg-2Ag for an intramedullary nail to fix the long bone fractures. The implants had a diameter of 0.8 mm, and they were implanted in the femoral zone of mice. The Mg-2Ag alloy side effects were investigated by mice body weight measurements and analysis of liver, kidney, spleen, and muscle. Taking into account the overall well-being of the mice, no histological abnormalities were reported. During the fracture repair process, the osteoblast function and bone formation were improved. In addition, a decrease in the osteoclast activity and bone resorption, which leads to prolific callus generation during fracture healing, was noticed.

Relatively recent, some researchers proposed Mg alloys as a raw material for bone engineering scaffolds production [9,61,127,128]. Chen et al. [127] suggested a novel open-porous magnesium scaffold with controllable microstructures for bone regeneration made from 99.9% Mg ingots by the titanium wire space holder method. Using HF solution, the Ti wires were removed. Scaffolds of 3 mm diameter and 5 mm length were implanted into the rabbit femur condyle. The larger porous configuration presents a moderate inflammatory response. After a longer time, it was found that this scaffold leads to higher bone mass generation, due to an increased exchange of body fluids, vascularization, and up-regulated collagen type I and osteopontin (OPN) expression. Liu et al. [128] have prepared microarc surface-treated hollow cylindrical scaffolds from pure Mg. These implants were placed in the right femoral condyle of NZW rabbits. The porous Mg scaffolds present good degradation and osteogenesis, with normal liver and kidney function, indicating good biological compatibility.

## 5. Clinical Translation of Mg-Based Materials for Temporary Implants’ Manufacture

Implants manufactured from Mg-based alloys have a great potential to treat broken bones because their mechanical strength is higher than that of polymeric substances and, also, because the degraded Mg particles are dissolved in the body fluids. Macrophage cells can have an influence on the excess quantity of degraded particles.

Mg-based alloys were tested in orthopedic surgery starting from 1906 when a tibia fracture fixation was made using a Mg plate and iron screws. Then, in 1932, four cases of the supracondylar fracture using 99.9% Mg nails, followed by a transdiaphyseal humerus fracture with a Mg-Al-Zn plate and screws, were solved. In 1940, two cases of humerus fractures using Mg sheets were solved, and in 1948, 34 cases of pseudoarthrosis using Mg-Cd plates and screws were fixed.

After a long time, researchers focused again on the Mg-based alloys trying to take advantage of newly developed technologies in materials science and engineering as well as in biology and medicine. An important fact that stimulates the research in this field was the fact that some companies develop and sell worldwide biodegradable orthopedic implants made by Mg-based alloys. Syntellix AG (Hanover, Germany) using MAGNEZIX^®^ (Mg-Y-RE-Zr) and U&I Corporation (Gyeonggi, Korea) using RESOMET™ (Mg-Ca alloy) looks to be more involved in publications related to the clinical testing of orthopedic Mg-based implants.

From a clinical perspective, thirteen cases of symptomatic hallux valgus using the commercially sold MAGNEZIX^®^ (Mg-Y-RE-Zr) were treated in 2010. In addition, 23 cases of osteonecrosis of femoral head flap fixation were made using pure Mg, and 53 cases of distal radius fracture fixation based on RESOMET™ (Mg-Ca alloy) were treated in 2013 [48].

In Table 7, some MAGNEZIX^®^ implants used as fracture fixture devices at different surgical areas are presented.

The most used temporary devices in orthopedic surgery are bone screws, interference screws, pins, plates, nails, wires, and scaffolds [149,150,151,152,153,154,155]. Some orthopedic devices made using different biodegradable Mg-based alloys, which are FDA approved or experimentally tested by various research groups, are presented in Figure 9.

A pilot randomized and prospective clinical trial regardiwiteng biodegradable pure Mg screws used to treat osteonecrosis of the femoral head (ONFH) was conducted by Zhao et al. [63]. The pure Mg screws were 4 mm in diameter and 40 mm in length. In total, 48 patients were subjected to autologous vascularized bone grafting, which were divided into two groups: for the first one, Mg screws were implanted to fix the bone grafting (Figure 10), and for the control group, the patients received the surgical procedure without fixation. The study reported a follow-up analysis of 12 months, pointing out a good degradation rate of the Mg screws, a better stabilization of the bone flap, low to nonlocal gas formation, no tissue necrosis, and abnormal blood chemistry after postimplantation.

Yu et al. [157] reported clinical research, in which 19 patients with displaced femoral neck fracture were treated using vascularized iliac graft implantation combined with fracture fixation and pure Mg screws. By clinical and radiological investigations, it was proven that no patient has developed avascular necrosis of the femoral head after the surgery. It was concluded that pure Mg screws and vascularized iliac grafting are adequate for femoral neck fracture treatment in young adults, with good results and a low rate of complications. Chen et al. [158] presented a case report for traumatic femoral head necrosis treatment, using a pedicled bone flap and pure Mg screws. The nail used for internal fixation was removed, the necrotic bone tissue was cleaned, and the defect was treated with an iliac bone grafting, fixed with Mg screws. After 2 years of follow-up, the 17 years old patient (in 2019) had no significant progressive necrosis of the femoral head, and the diameter of the screw was reduced, showing an adequate Mg degradability process. By analyzing this case, it was concluded that the treatment was efficient, and the pure Mg screws helped the osteointegration of the graft.

The Mg-Y-RE-Zr alloy is considered the most advanced system in Mg-based alloys for medical devices, being in clinical use since 2013. Windhagen et al. [139] conducted the first prospective, randomized, and controlled clinical pilot study regarding biodegradable magnesium-based screws for hallux valgus surgery. There were implanted cannulated screws (shaft diameter of 2 mm and cannulation diameter of 1.3 mm) with two threads (with diameters of 3 and 4 mm). The study involved 26 patients, divided into two groups: one group received the Mg-Y-RE-Zr screws and the other group standard Ti screws. The six months follow-up analysis reported no allergic reactions, foreign body, and systemic inflammatory reactions, or osteolysis and complete bone healing were observed. In this study, radiographic and clinical results, which show that the degradable Mg-based implants are equivalent to Ti screws in hallux valgus treatment (Figure 11), were performed and discussed. The hydrogen presence was noticed in X-rays images, but after 3 months, it completely disappeared.

Choo et al. [135] showed their results on 24 patients with hallux valgus deformity were surgically treated by scarf osteotomy method, applied to the first metatarsal, using Mg-Y-RE-Zr screws. A follow-up period of 12 months was enough to obtain radiological outcomes, functional scores, and complication profiles and to compare these results to those obtained in the case of a control group consisting of 69 patients with inserted Ti alloy screws. In this clinical study, some complications, such as superficial cellulitis and neuropathic operative site pain, were recorded. The superficial cellulitis was solved after 1 week of antibiotics treatment. In the case of one patient, the implant was removed because of a complex regional pain syndrome. For the MAGNEZIX^®^ screw patients, a delayed wound healing was observed, but as an overall impression, the Mg alloys’ implants proved to be comparable, taking into account their outcomes, to the Ti screws’ performance.

Klauser et al. [137] performed a clinical study on 100 patients with hallux valgus deformity using MAGNEZIX^®^ screw as implants and Chevron and Youngswick osteotomies as surgical techniques. The results were analyzed compared to those recorded in the case of 100 patients with Ti screws with similar surgical techniques. The investigated cohort showed no important delayed wound healing for Mg screw treatment. In addition, in this paper, it was established that Mg screws produce a similar effect as Ti implants.

May et al. [140] analyzed a group of 48 patients with medial malleolar fractures, from which 23 people were treated with Mg-Y-RE-Zr compression screws and 25 patients with conventional Ti screws. A follow-up time of 1 year was taken into consideration, and it was concluded that Mg and Ti screws fixed in a similar fashion the medial malleolar fractures. The same radiological and functional outcomes were obtained. A higher rate of implant removal in the case of the Ti control group was noticed, and for the bioabsorbable Mg screws, it was concluded that they are a more favorable fracture fixation option, because a secondary implantable screw removal surgery was not needed. In [141], a cohort of 22 patients underwent a medial malleolar osteotomy for treatment of osteochondral lesions of the talus. From this group, 11 patients had a defect fixation with Mg-Y-RE-Zr screws. It was noticed that after a 1-year follow-up, a complete union of the osteotomy was obtained for all 22 patients. One patient from the control group reported irritation and pain, and an implant removal surgery was performed. The main conclusion was that no important complications were observed and bioabsorbable Mg screws are a viable alternative to the conventional Ti screws.

Leonhardt et al. [143] made a retrospective observational study consisting of six patients, which suffered a mandible fracture and were treated with Mg-Y-RE-Zr headless compression screws. All the patients exhibited a restored function of the temporomandibular joint and an important improvement in mouth opening. The osseous remodeling of the mandibular condyle was shown by radiological methods, and a few radiolucencies indicated the Mg screw presence. One screw penetration through the condylar surface was reported, but the implant removal was not necessary, due to Mg-based alloy biodegradability.

Aktan et al. [21] reported a fixation system for small osteochondral fragments in a comminuted distal humerus fracture based on Mg screws. Headless compression screws and K wires in the case of a 50-year-old patient (in 2018) were used. A posterior surgical approach with olecranon osteotomy was employed, and the articular surface was reduced and fixed with two 2.7 mm diameter Mg-Y-RE-Zr bioabsorbable screws, and the lateral column was stabilized with an anatomic Ti lateral column plate. The Mg and Ti were not in direct contact (Figure 12). In the first four months, it was observed that a limited and insignificant gas quantity was present around the Mg screws, but the quality of joint surface reduction was smooth and healthy.

Grieve et al. [146] analyzed a case series of six patients, which were treated with 3.2 mm MAGNEZIX^®^ screw, three scaphoid fixations, and three intercarpal fusions. A follow-up time between 6 and 18 months was applied for the investigated cohort. Medical images showed the healing progress at 6 and 12 weeks. Lucency and gas formation in the implant neighborhood were noticed, and no systemic complications were reported. The Mg screws proved to be efficient in carpus fracture healing.

Meier and Panzica [147] described five cases of patients with acute scaphoid fractures, which were stabilized with Mg-Y-RE-Zr compression screws. Clinical and radiological follow-up at 6 weeks, 3 and 6 months, and 1 year after surgery were made. A good wrist score was recorded for all the patients, but the X-ray images showed resorption cysts in the case of three patients. After 6 months, the fracture was healed, and the patients properly used their hands.

Gigante et al. [22] presented a case series regarding the intercondylar eminence fracture treated with MAGNEZIX^®^ screws. From the analyzed cohort, three patients were treated using internal fixation with Mg-based screws. A follow-up of 12 months, with an intermediary investigation step made at 6 months was chosen. After 6 months, the Mg implants appear completely resorbed, and after 12 months, the devices were replaced with new bone. It was concluded that tibial spine avulsion fracture can be efficiently treated with arthroscopic reduction and internal fixation using MAGNEZIX^®^ screws, and the excellent functional recovery of the affected limb with no supplementary complications was obtained.

Wichelhaus et al. [149] studied a case of implant failure in partial wrist fusion in the case of a 42-year-old patient. The implant failure was due to osteolytic seams around screws with cystic formations, detected in the trapezoid, scaphoid, and trapezium. Revision surgery was needed, and large gaps were found near the Mg screws. The screw threads changed their shape, and the tissue around the implants became blackish. In 2014, the patient presents to the clinic with symptomatic periscaphoid osteoarthritis. A conclusion was that due to early degradation of screw and loss of mechanical properties, it was impossible to produce a union and proper osteolysis of the three carpal bones.

An improved alloy configuration is presented by Lee et al. [159], where a ternary Mg-5Ca-1Zn alloy was used to manufacture screw implants of 2.3 mm in diameter and 14 mm in length to fix a distal radius fracture. The clinical study consisted of 53 patients’ investigation, showing a good healing process that permits cortical bone formation and a small diameter of the Mg-Ca-Zn screws after surgical implantation. A bone fracture after 6 and 12 months after implantation was not reported. What is more, the patients had no pain and the imagistic investigations showed a complete healing process of the bone, simultaneously with total resorption of Mg-Ca-Zn screws. The most important conclusion of this study was that a Mg-based alloy, which does not contain potentially harmful chemical elements, such as Al and RE, is very suitable for small bone fracture healing. In Figure 13, the X-ray images made after a 1-year follow-up in the case of a 29-year-old female patient, are presented. In this case, the scaphoid fracture was fixed with two conventional stainless-steel pins, and the distal radius fixation was made using a Mg-5Ca-1Zn screw. After 6 months, the radius fracture was completely healed, and a very small radiolucent zone in the screw insertion site was visible.

There are important challenges for Mg-based implants if they are used in weight-bearing skeletal zones [160,161]. From a future perspective, the degradation of the Mg alloy orthopedic devices has to be optimized [162,163,164]. A cavity formation, surrounded by fibrous tissue near the implant site after its degradation, was observed; therefore, solutions such as inorganic or organic coatings applied on the Mg-based alloy surface should be considered in order to accurately control the degradation rate of the material. Mg-based alloys exhibit insufficient mechanical strength for load-bearing applications, and a new implant design must be taken into account for these areas [164]. Sometimes, the breakage of the screw head can occur during the surgical procedures, and these unwanted events show that Mg implants must meet the demand of higher torque for future use in humans.

Scaffolds made from Mg-based alloys associated with biodegradable polymers or ceramics, obtained by various techniques, enhance the healing of bone defects with the support of Mg ions, which induce and accelerate new bone formation [165].

Another interesting option for the treatment of bone fractures is the development of some hybrid systems made by titanium plates fixed with titanium and magnesium screws [166]. A combination between Mg-based implants and inert metals (Figure 14) can expand the applications of implants to the low and high bearing skeletal zones, and more Mg-based alloys can be approved for clinical use in many orthopedics areas.

## 6. Conclusions

Mg-based alloys have become in the last years a very important biomaterial category class in the framework of biodegradable metals. Their biodegradability in the human body environment makes them suitable for manufacturing different orthopedic temporary implants. Designing the new Mg-based alloys for medical applications, by bringing new alloying elements, with careful attention played to their systemic toxicity characteristics, is very important because these elements improve the mechanical properties and increase the corrosion resistance, due to their effects on Mg-based alloys microstructure. In addition, special attention must be given to all metallurgical aspects because these could influence the final functional properties by the triangle “chemical composition-processing-properties”.

Different research groups demonstrated that the initial difficulties with in vitro testing appear to be surpassed, and the testing procedures and the mediums used are quite well accepted.

Animal testing is widely presented in the literature, and the main species used for in vivo experiments are mice, rats, rabbits, dogs, goats, and mini pigs. The advantages and drawbacks of these animal models were highlighted, and they can be correlated also with the in vitro tests.

The major advantage to the field is given by the presence on the market of some commercial implants used as screws and pins, as well as other experimental orthopedic implants plate-screw type or experimental scaffolds which are in the preclinical tests stage. An example consists of the implants that are currently commercialized, such as MAGNEZIX^®^ (Mg-Y-RE-Zr) or RESOMET™ (Mg-Ca) for orthopedic surgery. For these devices, a secondary surgery is not needed, because it was proven that in time, they entirely dissolve in human fluids, and their corrosion products are not harmful even in the case of hydrogen emission, which is eliminated after some time.

Future research must be concentrated on the direction of alloys with a low degradation rate and an improved mechanical strength, in order to solve load-bearing zone fractures. In addition, new alloying elements must be searched, for increased biocompatibility of the alloys. New designs for orthopedic implants are possible to be developed in the near future, especially for foot and ankle surgery, if the researchers better correlate the clinical needs with each Mg-based alloy biofunctional properties. This is because, in the case of biodegradable Mg-based alloys, we cannot conclude that a universally accepted alloy for any orthopedic applications exists. Further research that includes human studies is indicated for each newly developed implant.

Apart from the previously mentioned future directions, we consider that, soon, it will be possible to move forward, in order to escape from the paradigm that Mg-based alloys are good only for small implants, to the implants or hybrid systems for large bone defects.

In addition, the Mg-based alloys have a huge potential in the regenerative medicine field, due to the osteogenic properties of magnesium and its potential use for osseous defects caused by degenerative diseases or bone cancer repairing, in association with biodegradable ceramics and polymers.

## Figures and Tables

**Figure 1 materials-15-01148-f001:**
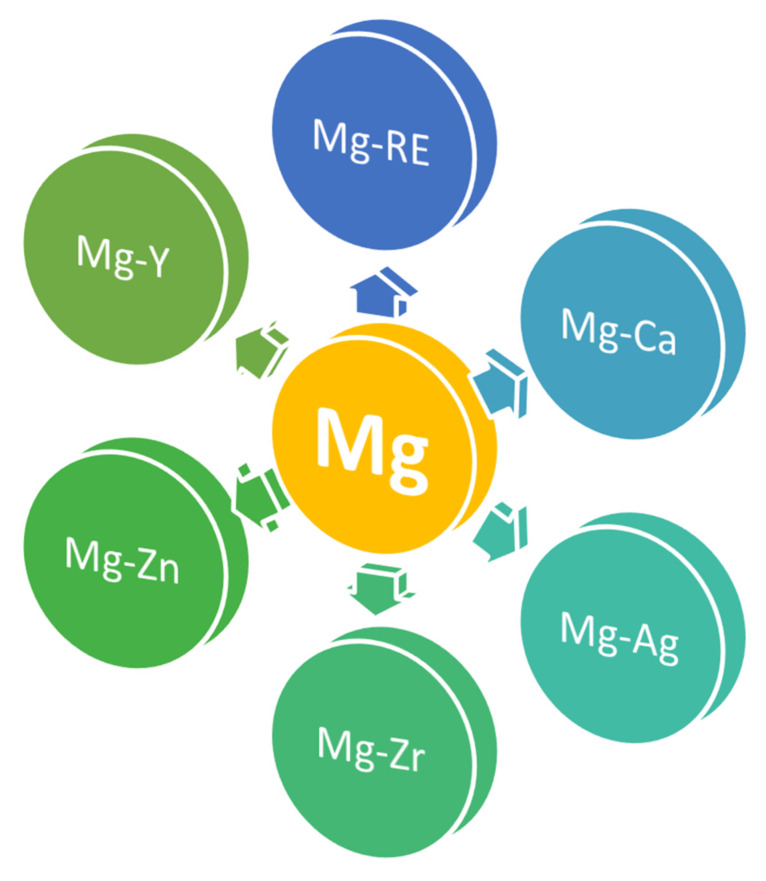
The main six Mg-based binary alloys for orthopedical applications.

**Figure 2 materials-15-01148-f002:**
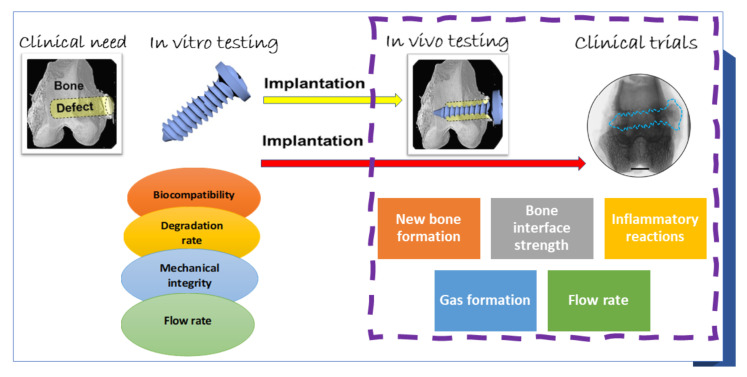
Methods used for the evaluation of the biodegradable Mg alloys for temporary orthopedic implants.

**Figure 3 materials-15-01148-f003:**
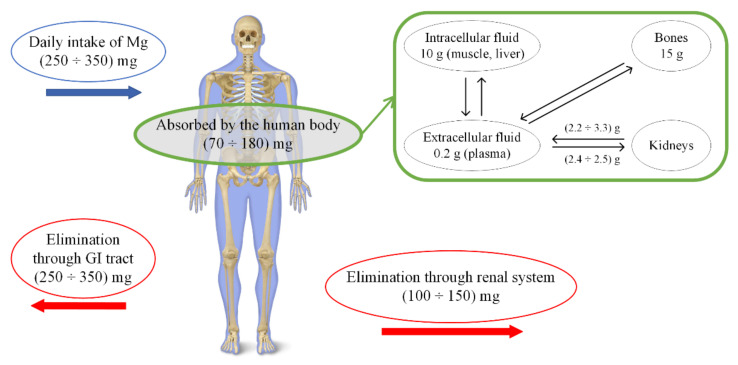
Absorption phenomenon and excretion equilibrium of Mg in the human body system.

**Figure 4 materials-15-01148-f004:**
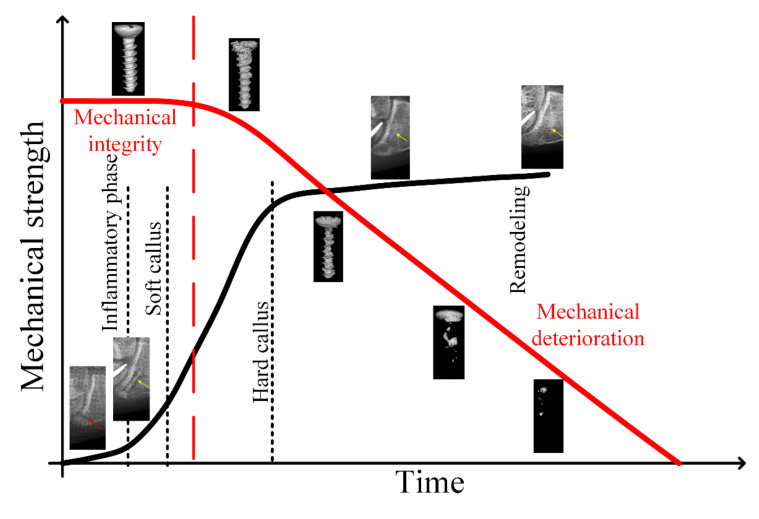
Degradation behavior of Mg-based temporary implants in bone fracture healing process, in ideal conditions (adapted after [50]).

**Figure 5 materials-15-01148-f005:**
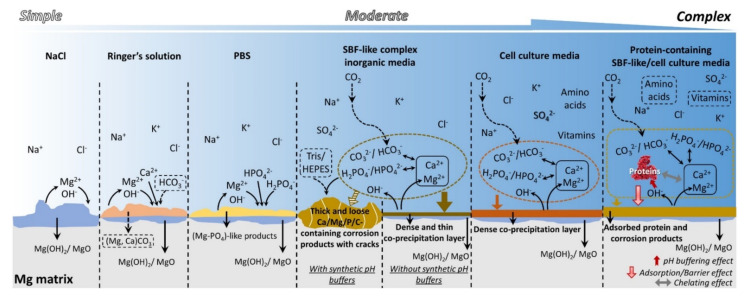
Schematic illustration of the corrosion behavior of Mg in the commonly used media [75].

**Figure 6 materials-15-01148-f006:**
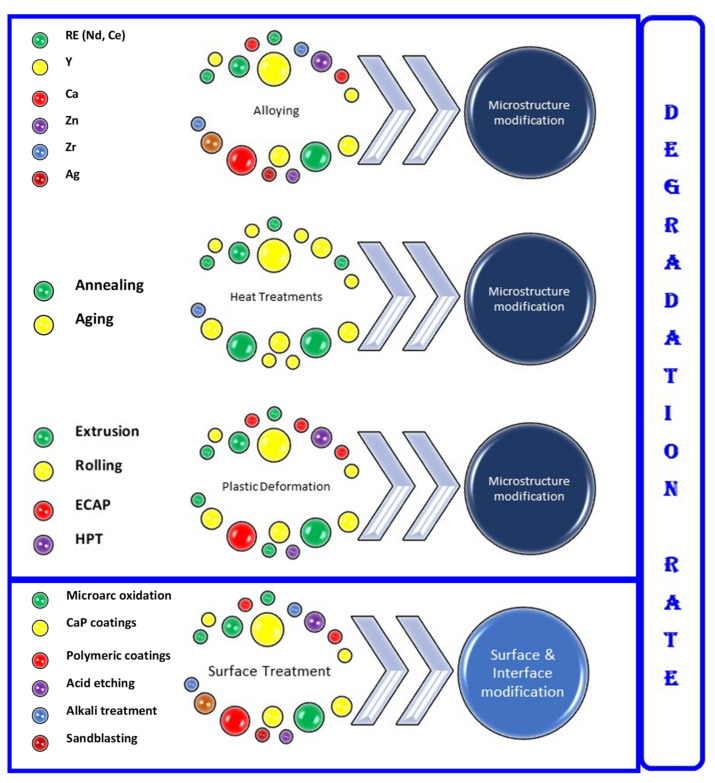
Schematic illustration of the mechanisms to control the degradation rate of Mg-based alloys.

**Figure 7 materials-15-01148-f007:**
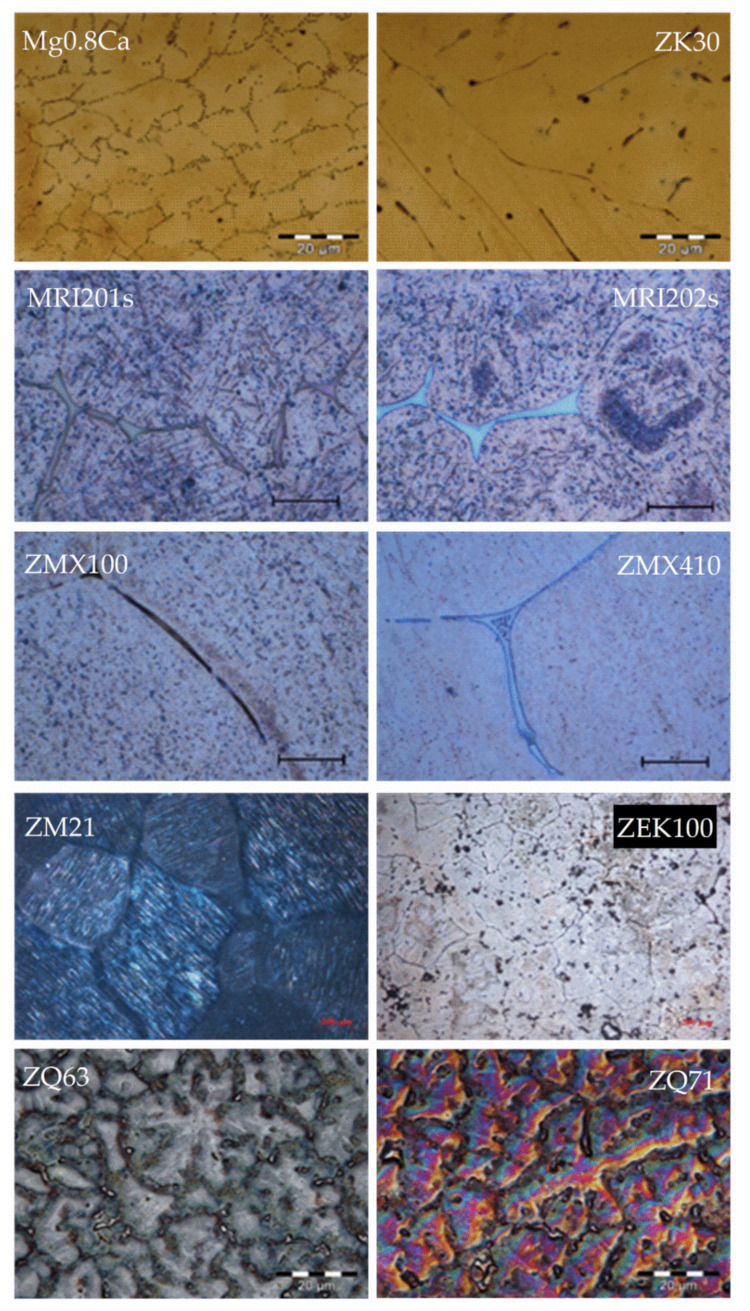
Optical microscopy images of some magnesium alloys for biomedical application.

**Figure 8 materials-15-01148-f008:**
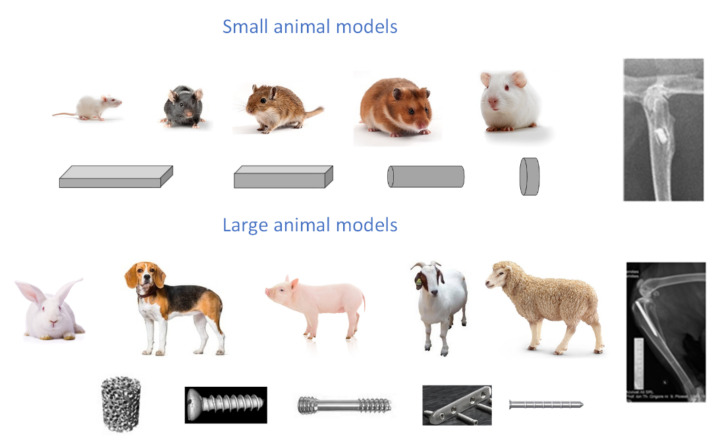
Small and large animal models and associated tested geometries.

**Figure 9 materials-15-01148-f009:**
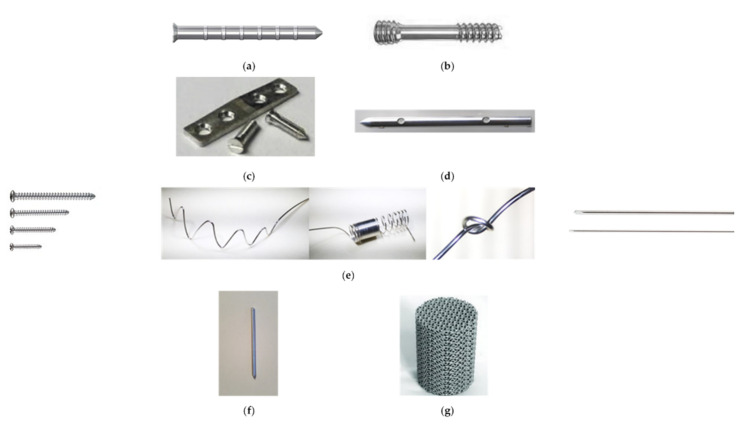
Examples of bone fixation devices made from Mg and Mg-based alloys: (**a**) pins from MAGNEZIX^®^ [152] (courtesy of Synthellix AG, Hannover, Germany); (**b**) compression screw from MAGNEZIX^®^ [156] (courtesy of Synthellix AG, Hannover, Germany); (**c**) magnesium plate and screw for fracture fixture [43], (**d**) intramedullary nail made from magnesium [43], (**e**) screw and wire from RESOMET^®^ [154], (**f**) pin [129], and (**g**) scaffold [127].

**Figure 10 materials-15-01148-f010:**
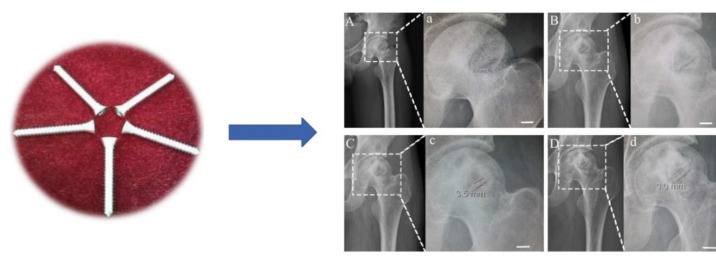
Mg-based screws used for bone flap fixation (shaft diameter = 4 mm and length = 40 mm). X-ray images of the femoral head in which Mg screws were implanted at 1 (**A**), 3 (**B**), 6 (**C**), and 12 (**D**) months after surgery. (**a**–**d**) Details of surgical zones taken for screw diameter measurement (scale bar is 10 mm) [63].

**Figure 11 materials-15-01148-f011:**
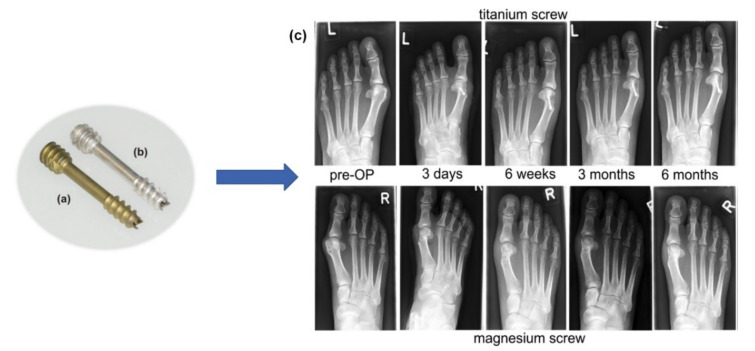
(**a**) The titanium screw from Koenigsee Implantate GmbH. (**b**) MAGNEZIX^®^ Compression screw from Syntellix AG. (**c**) Preoperative and postoperative X-rays images of a hallux valgus deformity. Correction was performed using a Chevron osteotomy [139].

**Figure 12 materials-15-01148-f012:**
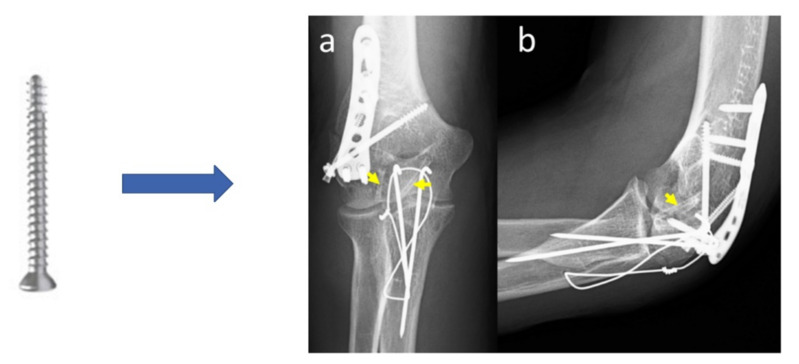
Patient radiographs (**a**) anteroposterior- and (**b**) lateral- elbow X-ray images taken after the operation. Yellow arrows marked the places, where Mg screws are used [21].

**Figure 13 materials-15-01148-f013:**
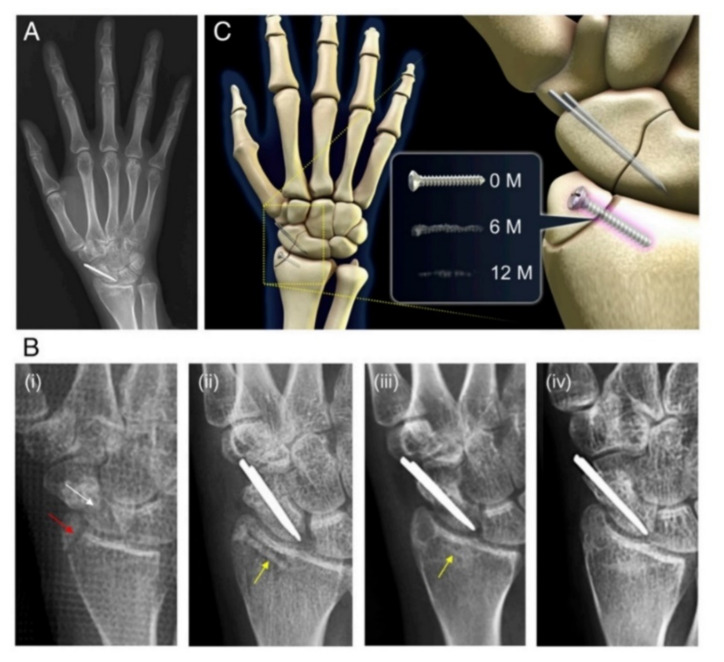
Clinical observation of complete screw degradation and bone healing. (**A**) 1 year follow up; (**B**) radiographs, (**i**) distal radius fracture, (**ii**) implantation site immediately after surgery, (**iii**) 6 months after surgery, and (**iv**) 1 year follow up situation; and (**C**) schematic diagram of the case [159].

**Figure 14 materials-15-01148-f014:**
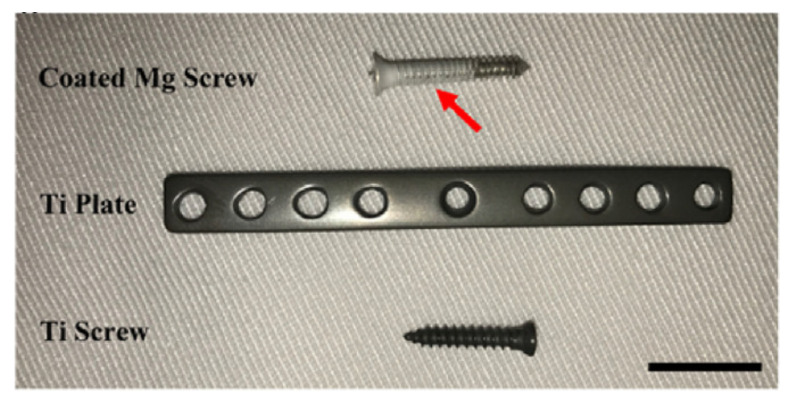
Hybrid system used for mandible fracture fixation. The system is composed of a Ti plate (2 mm × 50 mm), a Mg screw covered with PLA coating, and a Ti screw (the scale bar is 10 mm) [166].

**Table 1 materials-15-01148-t001:** Temporary fixation devices for different fracture types in human and animal models.

Fracture Type	Implant Type	Mg-Based Alloy	Animal Model/Human	Ref.
Skull fracture	Four-hole plates (1 mm thickness, 22 mm in length) and cortical bone screws (2 mm in diameter, 5 mm in length)	WE43/Mg-Y-RE-Zr (Syntellix AG)	Miniature pig	[20]
Comminuted distal humerus fracture	2.7 mm diameter screws	Mg-Y-RE-Zr (MAGNEZIX^®^, Syntellix AG)	Human male 50 years old (2018)	[21]
Tibial spine fracture	Magnesium screws	Mg-Y-RE-Zr (MAGNEZIX^®^, Syntellix AG)	Human female 20 years old (2018)	[22]
Femoral fracture in osteoporotic bone	Intramedullary nail	Mg-Nd-Zn-Zr (JDBM) integrated with the anti-catabolic drug zoledronic acid (ZA)	Rat	[23]
Elbow fracture	2.7 mm diameter screws	Mg-Y-RE-Zr (MAGNEZIX^®^, Syntellix AG)	Human female 48 years old (2020)	[24]
Osteotomized bone in sheep	screw	Mg-Zn-Ca (ZX00)	Sheep	[25]
Ulna fracture	Plates with an area of 20 mm × 4.5 mm, thickness of 1–1.5 mm, screws 7 mm in length,	Pure Mg (99.9%)	White rabbit	[26]
Scaphoid fracture	Magnesium-based headless Herbert screw	Mg-Y-RE-Zr/WE43	Human (190 patients)	[27]
Isolated lateral malleolar fracture	Cannulated headless compression screws	Mg-Y-RE-Zr	Human female 19 years old (2018)	[28]
Patella fracture	Fixation pin with a diameter of 1 mm	Pure Mg (99.9%)	New Zealand white rabbits	[29]

**Table 2 materials-15-01148-t002:** Physiological and toxicological characteristics of the alloying elements in Mg-based alloys [59].

Alloying Element	Physiological and Toxicological Characteristics
Magnesium (Mg)	The normal blood quantity of Mg ranges between 0.73 and 1.06 mmol/L. It has a benefic effect on metabolism, cells proliferation, and protein synthesis. It regulates the activity of about 350 different proteins, and it stabilizes the DNA and RNA. It has a long-term influence on cellular reactions.
Calcium (Ca)	The normal blood quantity of Ca ranges between 0.919 and 0.993 mg/L. It is the most abundant mineral material in the human body and is deposed in bones. The skeletal, renal, and intestinal homeostases regulate the Ca quantity.
Zinc (Zn)	The normal blood quantity of Zn is between 12.4 and 17.4 μmol/L. It is a trace element that is essential for the immune system. It is considered an enzymatic agent for bones and cartilages. High concentrations can exhibit neurotoxic effects.
Manganese (Mn)	The normal blood quantity of Mn must be lower than 0.8 μg/L. It is an essential trace element. Mn plays an important role in the metabolic circuit of lipids, amino acids, and carbohydrates. It has an influence on the immune system, bone growth, and blood coagulation. At high concentrations, it can exhibit neurotoxic effects.
Rare earth (RE)	A lot of rare earths exhibit anticancer properties.
Strontium (Sr)	The normal blood quantity of Sr should be equal to 0.17 mg (total). Notably, 99% of total Sr is located in bones, and it proves a metabolic effect on bone and stimulates new bone formation. Sr in high doses results in hypocalcemia or skeletal unwanted effects.
Zirconium (Zr)	The normal blood quantity of Zr must be lower than 0.250 mg (total). Zr shows low ionic toxicity and good biocompatibility. Zr accumulates in the bone and nervous system.

**Table 3 materials-15-01148-t003:** Physiological and toxicological characteristics of the impurities in Mg-based alloys [59].

Impurity	Physiological and Toxicological Characteristics
Nickel (Ni)	The normal blood quantity of Ni ranges between 0.05 and 0.23 μg/L. It manifests allergenic properties, and it can induce metal sensitization. It has a strong carcinogenic and genotoxic effect.
Beryllium (Be)	The toxic dose is considered higher than 2 μg/m^3^. It has carcinogenic potential and induces metal sensitization.
Iron (Fe)	The normal blood quantity of Fe is between 5 and 17.6 g/L. It is regulated and deposited through human body metabolism.
Copper (Cu)	The normal blood quantity of Cu ranges between 74 and 131 μmol/L.

**Table 4 materials-15-01148-t004:** Chemical composition of some biodegradable magnesium alloys for medical applications.

Magnesium Alloy	Element (wt.%)										
**ASTM**	**Al**	**Zn**	**Mn**	**Re ^a^**	**Zr**	**Y**	**Ca**	**Ag**	**Nd**	**Cu**	**Mg**
AZ31A	2.4	0.50	0.15	-	-	-	-	-	-	-	bal
AZ31B	2.4	0.50	0.05	-	-	-	-	-	-	-	bal
AZ61A	5.5	0.50	0.15	-	-	-	-	-	-	-	bal
AZ80A	7.8	0.20	0.12	-	-	-	-	-	-	-	bal
MC1	-	-	-	-	-	-	0.80	-	-	-	bal
ZK60	-	8.00	-	-	-	1.50	-	-	-	-	bal
ZM21	-	2.00	1.16	-	-	0.16	-	-	-	-	bal
EW10	-	-	-	-	0.50	0.50	-	-	1.20	-	bal
ZEK 100	-	1.00	-	0.12	0.20	0.17	-	-	-	-	bal
ZMX410	-	4.30	0.62	-	-	-	0.30	-	-	-	bal
ZE41	-	4.20	-	1.30	0.40	-	-	-	-	-	bal
ZC71A	-	6.0	0.5	-	-	-	-	-	-	1.00	bal
WE54A	-	-	-	1.50	-	4.75	-	-	-	-	bal
WE43A	-	-	-	2.40	-	3.70	-	-	-	-	bal
ZQ71	-	7.2	-	-	1.30	0.20	-	1.50	-	-	bal
ZQ63	-	6.4	-	-	1.00	0.16	-	2.50	-	-	bal
MRI 201s	-	0.3	-	-	0.60	2.10	-	-	3.20	-	bal
MRI 202s	-	0.3	-	-	0.40	0.21	-	-	3.10	-	bal
ZMX 410	-	4.3	0.62	-	-	-	0.30	-	-	-	bal
ZMX 100	-	1.3	0.51	-	0.03	-	0.38	-	-	-	bal

^a^: It is Rare Earth Element (other than Y and Nd). Ce and La.

**Table 5 materials-15-01148-t005:** Mechanical properties of as-cast Mg-based alloys that do not contain Al.

Alloy Composition (wt.%)	Yield Strength (MPa)	Tensile Strength (MPa)	Elongation (%)	Hardness	Ref.
Mg-4.71Y-4.58Gd-0.31Zr.	255	330	15	1200 Hv	[66]
EW10 (Mg-1.2Nd-0.5Y-0.5Zr)	77 ± 4	175 ± 11	12 ± 3	-	[67]
EW10 + 0.4Ca	74 ± 5	135 ± 11	5 ± 1	-	[67]
ZK60 (Mg-8Zn-1.5Y)	390	445	8.3	69 Hv	[68]
WE43 (Mg-4.38Y-2.72Nd-1.1Gd-0.56Zr)	145 ± 16	204 ± 6	6.9 ± 0.5	85 Hv	[69]
Mg- 5Zn	68 ± 1.5	185 ± 5	9.2 ± 0.5	-	[57]
Mg-4Zn-0.2Ca	58.1 ± 1	255 ± 5	17.5 ± 1	-	[70]
Mg–Zn -Mn	78 ± 2	175 ± 3	12	68 Hv	[71]
Mg- 4Zn-0.5Ca-0.16Mn	175	180	0.2	70 Hv	[72]
Mg-2Zr-2Sr	80	290	15	-	[73]
JDBM (Mg-3.13Nd-0.16Zn-0.41Zr)	189 ± 2	243 ± 3	21 ± 0.9	-	[74]

**Table 6 materials-15-01148-t006:** Examples with different animal models and Mg-based implants used in the studies.

Animal Model	Age/Weight	Fracture Site	Mg-Based Implant Type	Implant Type	Follow-Up	Ref.
Mouse	10 weeks	femur	Mg-2Ag	Intramedullary nail	133 days	[121]
Mouse	3 months	femur	Mg-2Sr	Rods	30 days	[117]
Albino rat	8 weeks	femur	Mg-1Ca-0.2Mn-0.6Zr	Intramedullary bar	180 days	[122]
Rat	-	femur cortical bone	Mg-1Zn-0.8Mn	Rod implant	182 days	[123]
Sprague-Dawley (SD) rat	250 g	femur	Mg-4Y-2Zn-1Zr-0.6Ca (WZ42)	Intramedullary pin and wire	98 days	[124]
Sprague-Dawley (SD) rat	9 months	femur	Mg-3Nd-0.2Zn-0.4Zr (JDBM)	Intramedullary pin	84 days	[23]
Sprague-Dawley (SD) rat	8 weeks/220 g	femur	Mg-0.8Ca Mg-0.8Ca-5Zn-1.5Ag Mg-0.8Ca-5Zn-2.5Ag	Cylindrical samples	28 days	[111]
Dunkin Hartley guinea pig	658 g	femur	WE43	Rods	126 days	[125]
New Zealand White (NZW) rabbit	19 weeks	ulna	99.9% Mg	Plate and screw	28 days	[126]
New Zealand White (NZW) rabbit	6 months/2.5 kg	femur	Mg-5Zr Mg-1Zr-2Sr Mg-2Zr-5Sr	Cylindrical samples	90 days	[73]
New Zealand White (NZW) rabbit	3.87 kg	lateral epicondyle femur	High purity Mg	Porous scaffold	112 days	[127]
New Zealand White (NZW) rabbit	6 months	right femoral condyle	High purity Mg	4-hole cylindrical scaffold	84 days	[128]
Oryctolagus Cuniculus rabbit	3.5 kg	femur	Mg-1Ca	Parallelepiped samples	42 days	[129]
Beagle dog	1 year-10 kg	tibia	WE43	Screw	84 days	[130]
Goat	-	femur	JDBM	Screw	548 days	[131]
Sheep	1 months	tibia	Mg-0.45Zn-0.45Ca (ZX00)	Screw	84 days	[25]
Mini pig	14 months, 50 kg	frontal bone	WE43	Osteosynthesis plates and screws	210 days	[20]
Miniature mini pig	30–36 months, 53 kg	frontal bone	WE43	Plate with plasma electrolytic coating	168 days	[132]
Mini pig	53 kg	mandibular bone	modified WE43	Based rivet screws	168 days	[133]

**Table 7 materials-15-01148-t007:** MAGNEZIX^®^ implants for fracture at different surgical sites.

Clinical Needs	Patients	Clinical Outcomes	Follow-Up	Complications	Ref.
Hallux valgus	24	Similar functional outcomes with titanium screw used as control group	12 months	None	[135]
	13	Similar functional outcomes with titanium screw used as control group	3 years	None	[136]
	100	Similar functional outcomes with titanium screw used as control group	12.2 weeks	Soft tissue irritation, delayed wound healing, screw fracture	[137]
	16	Excellent	17.6 months	Prolonged swelling	[138]
	13	Both groups (Mg and Ti screws) were similar regarding the functional outcomes	6 months	None	[139]
Malleolar fracture and osteotomy	23	Similar functional outcomes with titanium screw used as control group	1 year	None	[140]
	12	Similar to control titanium screw group	1 year	Pain and irritation	[141]
	11	Excellent	17 months	None	[142]
Mandible fracture	6	Improvement in mouth opening, left and right laterotrusion and protrusion distance	1 year	-	[143]
	5	Excellent with good occlusion	3 months	One screw fracture, revised with Mg screw	[144]
Humeral fracture (elbow)	1	Excellent	4 months	None	[21]
	1	Excellent	24 months	None	[145]
Carpus	6	Good results	6–18 months	None	[146]
	5	Excellent	24 months	Extensive resorption cysts in the case of 3 patients	[147]
Knee intercondylar tibial eminence fracture	3	Excellent with new bone formation observed at the end of the follow-up time	12 months	None	[22]
Distal radius fractures	2	Excellent	27 months	None	[148]
	1	Poor	6 weeks	Revision following loosening and backing out of the screw, osteolysis and pain	[149]

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
