# Peer review of "Magnesium-Based Alloys Used in Orthopedic Surgery"

_materials, 2022, doi:10.3390/ma15031148_

Round 1

Reviewer 1 Report

Review report attached.

Author Response

Dear reviewer,

Thank you for the time spent to analyse our manuscript. We also thank you for all the suggestion. As you will notice, the entire manuscript was revised.

English is not satisfactory. Publication is not recommended unless intensive amendments on English is done. Most of the time, I do not get the meaning of the content especially on Mg alloys sections, technical and scientific aspects. One of the authors with materials science background needs to check and amend it very seriously.

Thank you for the observation. The new version of the manuscript was carefully amended in terms of formulations and scientific aspects.

  1. Starting in the abstract, long sentences with many grammar mistakes were

We have changed the entire manuscript, replacing long sentences and correcting the grammar mistakes.

  1. Figures 4 and 5 are not necessary as there was no explanation on those figures are Just showing the figures does not add up any value for the review paper. Remove both figures. Again, there were grammar errors too.

Thank you for your suggestion. We removed figure 5 and inserted new figures, in order make this review manuscript more useful for readers (see Figure 2, Figure 5 and Figure 6)

  1. Examples of sentences which I do not get the meaning due to grammar errors; and contain wrong technical/scientific descriptions

183: Mg2Ca are formed having as results the creep resistance increase, by fixing the grain limit

184: Using a higher amount of Ca (> 1 wt%) can cause problems with hot casting.

189-190: Rare earths (RE) are introduced by using pre-alloys, during the development of Mg alloys, having an important contribution on the mechanical strength of the material

208-210: The main methods through the Mg powders are made are atomization of molten metal, electrolysis, evaporation-condensation and mechanical crushing.

212-213: A technology developed in VAMI consist of atomization of the melt Mg with 212 the help of nitrogen jets, which contain an addition of 3 ± 0.1wt%.

225: Mg powders with a particle smaller than 1    m.

226: The machine that was involved, the so

258: The hydrogen gas, which result after the Mg corrosion has

260-262: It is still an actual challenge to develop such a material and to choose a proper production method, so the Mg-based alloy to exhibit ideal properties and to be used in orthopedic surgery without side effects and need of a secondary surgery [44,45].

271: foreign body reactions, which have as result a diminished life-expectancy period for 438-440: A fraction of intermetallic phase in the cast and extruded 438

Mg-RE alloys is due to high maximum solid solubility limit of rare earth and it provides good mechanical properties [78].

451: There were used fibroblast cells L-929 and NIH3T3 and murrine calvarial preosteoblasts MC3T3-E1.

483-485: 85]. The phase Mg97Zn1Y2 has a YS of 610 MPa and an elongation of 5% and it is present in an alloying system produce through powder metallurgy route [86].

608: the material with 5%Sr exhibits larger grain boundary zones and

557-559: The as-cast Mg-1Ca alloy has unsatisfactory mechanical properties so a hot rolling or a hot extrusion are necessary, because they promote a microstructure refinement.

601: If RE as holmium (Ho) is added intermetallic phases as MgHo3 and Mg2Ho, and together with Mg17Sr2 contribute

612-615: Hydrogen evolution test in the case of the investigated alloys points out the lowest hydrogen production rate in SBF for Mg-1Zr-2Sr and the highest value for Mg-5Zr-ySr. Mg- 1Zr-2Sr exhibits the best corrosion resistance in SBF solution

We thank the reviewer for pointing out all of these major errors. We have corrected the entire manuscript. We cannot point out the new lines, since the entire manuscript was completely reformulated. Thank you again.

4.    Some grammar errors

43: leaving bodies

130: Another biodegradable material is magnesium (Mg) which exhibit 177: Manganese is manly used to

181: act as a grain finishing agent

192: Other RE with limited solubility are

199: hot deformation the Mg-based alloy properties 200: The majority of the of the Mg based materials 202: Due to Mg high affinity for oxygen

204: to reduce de oxidation of Mg

206: a heat treatment or a hot deformation are applied 212: A technology developed in VAMI consist of

234: Granulated Mg can pe produced using

246-247: the most used technique, which help the doctors

437: in the commercially WE43 material 562: In [102] was put in evidence

582: they are manly

599: where concentration x and y are lower than

701: Usually there are involved animal models as rabbits, rats, miniature pigs 703:

706: taken into consideration the fact that

We thank the reviewer for pointing out all of these grammar errors. We have corrected the entire manuscript. We cannot point out the new lines, since the entire manuscript was completely reformulated. Thank you.

  1. Just like this In the case of hot rolling treatment the UTSseveral other sentences throughout the

We have analyzed the entire manuscript and have reformulated the wrong sentences throughout the entire manuscript.  

  1. The Mg-RE alloys exhibit a good degradation behavior   in several places in the manuscript. Any specific reason for doing it? If not, only use the same font and format.

Thank you for the observations. We changed the font and format in the suggested places.

Thank you again for your time and effort.

The authors.

Reviewer 2 Report

Mg-based alloys are very promising materials having high potential as orthopedical temporary implants. This detailed review paper covers the alloy development and manufacture technique in the introduction section in a comprehensive state of the art. In the following sections, important attributes for Mg-based alloys involved in orthopedic implants fabrication, the physiological and toxicological effects of each alloying elements, the mechanical properties of as-cast Mg-based alloys that does not contain Al, osteogenesis and angiogenesis of Mg are presented. The main biocompatible Mg-based alloys are also detailed concerning mechanical properties, degradation behavior and cytotoxicity tests. Representative cases and the alloy chemical composition, implant characteristics and “in vivo” behavior were highlighted in the animal testing. More importantly, clinical cases conducted to human use with detailed conclusions are included. Readers should be able to grasp the main progresses about the research of Mg-based alloys for degradable implants. It is suggested that the authors carefully check the figures presented in the review, such as Fig.1 to Fig.6. If these figures are not the works of authors, the corresponding references are recommended to be indicated in the descriptions for each specific figure.

Author Response

We thank the reviewer for the time spent and kind evaluation of our paper.

Regarding the mentioned figures:

  • figure 1 belongs entirely to the authors of the manuscript and
  • figure 6 is also created by the authors using animal images found on the web, that are not subject to copyrights requests.

Thank you again,

Authors

Reviewer 3 Report

I have carefully reviewed the manuscript entitled “Magnesium based alloys used in orthopedy” and my comments on the manuscript are as follows:

  1. The figures are not well prepared, and the quality is also very bad. I suggest authors include more figures and add some more (at least 3-4) which show some mechanisms and recent advancements.
  2. I feel that the characterization section should be more detailed and if possible separate.
  3. Many expressions are too long, and the author should focus on reducing the overall size as many basic things which are present in many previous articles are unnecessary. This type of discussion can be reduced so that the originality of the articles can be enhanced.
  4. Many of the references are too old. Please give an updated literature review as there are lots of good research papers published in the last 2 years. Please add more references.

Overall the manuscript is well structured and can be considered for publication if authors can revise the manuscript.

Author Response

Dear reviewer,

Thank you for the observations made to our manuscript.

We have uploaded the revised version.

I have carefully reviewed the manuscript entitled “Magnesium based alloys used in orthopedy” and my comments on the manuscript are as follows:

  1. The figures are not well prepared, and the quality is also very bad. I suggest authors include more figures and add some more (at least 3-4) which show some mechanisms and recent advancements.

Thank you for the suggestion. We have included more figures in the suggested chapters (current Figure 2, Figure 5 and Figure 6). We also removed, at the other reviewer suggestion one of the figures. The quality of figures is a consequence the pdf transformation, and we will upload in separate files the enhanced quality figures.

  1. I feel that the characterization section should be more detailed and if possible separate.

As a consequence of the previously mentioned modifications, the entire manuscript was changed in order to have a more readable sections.

  1. Many expressions are too long, and the author should focus on reducing the overall size as many basic things which are present in many previous articles are unnecessary. This type of discussion can be reduced so that the originality of the articles can be enhanced.

We have reviewed the entire manuscript, and we tried to reformulate all the long sentences. We reduced the overall size of the manuscript by removing the suggested unnecessary details. The entire manuscript (without references section) was reduced, globally, from 32 to 27 pages.

  1. Many of the references are too old. Please give an updated literature review as there are lots of good research papers published in the last 2 years. Please add more references.

Thank you for the suggestion. We replaced some old references and inserted more current ones. The entire reference part was revised.

e.g. of some inserted references

  • Bohlen, J.; Meyer, S.; Wiese, B.; Luthringer-Feyerabend, B.J.C.; Willumeit-Römer, R.; Letzig, D. Alloying and Processing Effects on the Microstructure, Mechanical Properties, and Degradation Behavior of Extruded Magnesium Alloys Containing Calcium, Cerium, or Silver. Materials (Basel) 2020, 13, E391, doi:10.3390/ma13020391.
  • Marukawa, E.; Tamai, M.; Takahashi, Y.; Hatakeyama, I.; Sato, M.; Higuchi, Y.; Kakidachi, H.; Taniguchi, H.; Sakamoto, T.; Honda, J.; et al. Comparison of Magnesium Alloys and Poly-l-Lactide Screws as Degradable Implants in a Canine Fracture Model: Comparison of Mg Alloys and PLLA Screws in Canine Fracture Model. J. Biomed. Mater. Res. 2016, 104, 1282–1289, doi:10.1002/jbm.b.33470
  • Kozakiewicz, M. Are Magnesium Screws Proper for Mandibular Condyle Head Osteosynthesis? Materials (Basel) 2020, 13, E2641, doi:10.3390/ma13112641.
  • Kania, A.; Nowosielski, R.; Gawlas-Mucha, A.; Babilas, R. Mechanical and Corrosion Properties of Mg-Based Alloys with Gd Addition. Materials (Basel) 2019, 12, E1775, doi:10.3390/ma12111775.
  • Rahman, M.; Dutta, N.K.; Roy Choudhury, N. Magnesium Alloys With Tunable Interfaces as Bone Implant Materials. Front Bioeng Biotechnol 2020, 8, 564, doi:10.3389/fbioe.2020.00564.
  • Huang, S.; Wang, B.; Zhang, X.; Lu, F.; Wang, Z.; Tian, S.; Li, D.; Yang, J.; Cao, F.; Cheng, L.; et al. High-Purity Weight-Bearing Magnesium Screw: Translational Application in the Healing of Femoral Neck Fracture. Biomaterials 2020, 238, 119829, doi:10.1016/j.biomaterials.2020.119829.
  • Zhu, J.; Jia, H. A Facile Method to Prepare a Superhydrophobic Magnesium Alloy Surface. Materials (Basel) 2020, 13, E4007, doi:10.3390/ma13184007.
  • Antoniac, I.V.; Antoniac, A.; Vasile, E.; Tecu, C.; Fosca, M.; Yankova, V.G.; Rau, J.V. In Vitro Characterization of Novel Nanostructured Collagen-Hydroxyapatite Composite Scaffolds Doped with Magnesium with Improved Biodegradation Rate for Hard Tissue Regeneration. Bioact Mater 2021, 6, 3383–3395, doi:10.1016/j.bioactmat.2021.02.030.

Overall the manuscript is well structured and can be considered for publication if authors can revise the manuscript.

Thank you for the comments.

Authors

Round 2

Reviewer 1 Report

Grammar check is still required.

Author Response

Dear reviewer,

Thank you for your observation.

We have uploaded a corrected version of the manuscript.

Authors

Reviewer 3 Report

The authors have revised the manuscript as per the suggestion. The Manuscript can be accepted for publication now.

Author Response

Dear reviewer,

Thank you for your effort.

Authors